# Sentinel Surveillance Contributes to Tracking Lyme Disease Spatiotemporal Risk Trends in Southern Quebec, Canada

**DOI:** 10.3390/pathogens11050531

**Published:** 2022-05-02

**Authors:** Camille Guillot, Catherine Bouchard, Kayla Buhler, Ariane Dumas, François Milord, Marion Ripoche, Roxane Pelletier, Patrick A. Leighton

**Affiliations:** 1Groupe de Recherche en Épidémiologie des Zoonoses et Santé Publique (GREZOSP), Departement of Pathology and Microbiology, Faculty of Veterinary Medicine, University of Montreal, Montreal, QC J2S 2M1, Canada; ariane.dumas@umontreal.ca (A.D.); patrick.a.leighton@umontreal.ca (P.A.L.); 2Faculty of Medicine and Health Sciences, University of Sherbrooke, Sherbrooke, QC J1H 5N4, Canada; francois.milord.med@ssss.gouv.qc.ca; 3Public Health Risk Sciences Divisions, National Microbiology Laboratory, Public Health Agency of Canada, St. Hyacinthe, QC J2S 2M1, Canada; catherine.bouchard@phac-aspc.gc.ca; 4Departement of Veterinary Microbiology, Western College of Veterinary Medicine, University of Saskatchewan, Saskatoon, SK S7N 5B4, Canada; kaylabuhler_1@hotmail.com; 5Public Health Directorate (Direction de Santé Publique), Centre Intégré de Santé et de Services Sociaux (CISSS) of Montérégie-Centre, Longueuil, QC J4K 2M3, Canada; 6Department of Biological Risks, Institut National de Santé Publique du Québec (INSPQ), Montreal, QC H2P 1E2, Canada; marion.ripoche@inspq.qc.ca (M.R.); roxane.pelletier@inspq.qc.ca (R.P.)

**Keywords:** sentinel surveillance, Lyme disease, tick-borne diseases

## Abstract

Lyme disease (LD) is a tick-borne disease which has been emerging in temperate areas in North America, Europe, and Asia. In Quebec, Canada, the number of human LD cases is increasing rapidly and thus surveillance of LD risk is a public health priority. In this study, we aimed to evaluate the ability of active sentinel surveillance to track spatiotemporal trends in LD risk. Using drag flannel data from 2015–2019, we calculated density of nymphal ticks (DON), an index of enzootic hazard, across the study region (southern Quebec). A Poisson regression model was used to explore the association between the enzootic hazard and LD risk (annual number of human cases) at the municipal level. Predictions from models were able to track both spatial and interannual variation in risk. Furthermore, a risk map produced by using model predictions closely matched the official risk map published by provincial public health authorities, which requires the use of complex criteria-based risk assessment. Our study shows that active sentinel surveillance in Quebec provides a sustainable system to follow spatiotemporal trends in LD risk. Such a network can support public health authorities in informing the public about LD risk within their region or municipality and this method could be extended to support Lyme disease risk assessment at the national level in Canada.

## 1. Introduction

Lyme disease (LD) is a tick-borne disease which has been emerging in temperate areas in North America, Europe, and Asia [1,2,3,4,5]. This emergence has been driven by the expansion of the geographical distribution of ixodid tick species (Acari: Ixodidae), which are vectors for LD pathogens [6,7]. In North America, ticks are dispersed over long distances by migratory birds from the US into Canada, and their range expansion has been further facilitated by a range of anthropogenic factors including climate change, land use modification and range expansion of their animal hosts [6,8].

As geographic range and abundance of ixodid ticks continue to increase, enzootic hazard of LD increases. In the context of LD, the enzootic hazard is defined as the potential source of harm for the disease derived by the enzootic cycling of *Borrelia* and can be measured as the density of nymphs (DON) in the environment [9]. As a direct consequence of increasing hazard, LD risk rises in human populations; there is an increased likelihood of the adverse effect (acquisition of LD) occurring due to the increased presence of a source of harm [10]. Thus, effective surveillance of enzootic hazard has the potential to provide valuable information about geographic and temporal variation in LD risk.

Common acarological surveillance methods which could be used to track the enzootic hazard include passive or active surveillance. Passive surveillance involves the submission of ticks to reference laboratories by medical or veterinary clinics from their patients. Passive surveillance data has been used as an early indicator of LD risk and has been shown to be closely correlated with the frequency of human LD cases [9,11]. However, analyzing high volumes of tick submissions can be very costly and resource intense, and may not remain feasible as tick populations continue to increase [12]. While data from passive surveillance can provide information on tick presence or absence, it is often not possible to know precisely where the tick was found or to estimate tick population density.

Active surveillance involves direct collection of ticks from their environment by a field agent [13]. For broader scale active surveillance initiatives, drag sampling is a method of choice [13]. A sheet of white flannel cloth is dragged upon the forest floor, and questing ticks will cling to the fabric. The tick specimens can be collected and identified, and the tick density can be calculated. For Ixodes ticks, the density of nymphal ticks (DON) is used as a measure of enzootic hazard, because nymphs are thought to represent the greatest risk of transmission of tickborne disease due to their small size, high relative abundance compared to adult ticks, and activity period that spans the spring and summer months [14]. While some associations between nymphal density and LD risk have been found [15,16,17], these are not always strongly correlated [9]. Furthermore, active surveillance requires intensive resources, both in the field and in the lab, and so surveying a large study zone can be a significant endeavor.

Sentinel surveillance, which involves repeated sampling of a select number of units from a population, has the potential to provide a feasible, sustainable surveillance system for LD by focusing active surveillance efforts on key locations and tracking this over time. Sentinel surveillance has been used historically, in many different infectious disease contexts, to maintain a surveillance system in various geographical regions at relatively low cost and with several logistical advantages in terms of data collection [18,19]. As tick populations are dispersed heterogeneously across space, in part due to complex ecological requirements [20], sentinel surveillance has the added benefit that, once appropriate sites are found, efforts can be redirected into sampling activities. Lastly, expected inter-annual variation in tick density [21], and resulting LD risk, can be measured more reliably if sampling sites are kept constant over time.

Among the ten provinces in Canada, Quebec (Figure 1) has the third-highest number of reported human LD cases; furthermore, in the last five years, the number of cases has more than tripled [22,23]. Quebec is the largest and second most populated province in Canada, with a population of almost 8.5 million citizens [24]. Most of the population resides in the south of the province, which coincides with the current LD emergence zone and highest *Ixodes scapularis* tick densities in the province. A sentinel surveillance network, composed of active surveillance sites sampled annually, was initiated by Quebec’s institute of public health (Institut National de Santé Publique, INSPQ) in 2015 [12]. After half a decade of surveillance, it is relevant to evaluate how well data collected through this surveillance system is tracking spatial and temporal variation in LD risk within the Quebec population. More broadly, sentinel surveillance has not been rigorously explored as a method for tracking spatial-temporal trends in tick-borne disease risk, and the Quebec surveillance program provides a unique context for assessing the strengths and limitations of this approach.

In this study, we aimed to understand how the spatio-temporal index of enzootic hazard obtained through active tick surveillance across a network of sentinel sites relates to Lyme disease risk in the human population, as measured by the annual incidence of human LD cases. We hypothesize that data from sentinel sites can be used to interpolate LD enzootic hazard across our study zone and subsequently can inform public health authorities about LD risk to human populations. We demonstrate that, even with limited sampling effort, sentinel surveillance has the capacity to reliably capture regional trends in LD risk over a large geographical area, in this case the full extent of the LD emergence zone in southern Quebec.

## 2. Results

### 2.1. Human Cases

Between 2015 and 2019, there were a total of 1062 human LD cases acquired in Quebec and reported in the ten RSS included in the study: 108 (2015), 125 (2016), 247 (2017), 210 (2018), 372 (2019). Estrie was the region with the highest reported number of human cases of Lyme disease per 100,000 population, followed by Montérégie (Figure 2).

### 2.2. Active Surveillance

A total of 207 sampling visits were conducted across 21 sentinel sites between 2015 and 2019: 38 (2015), 45 (2016), 43 (2017), 39 (2018), and 42 (2019). Average predicted nymph densities per region were calculated from raw data for each year of the study as described in the “Materials and Methods” section (Figure 3). The highest densities of nymphs were found in Montérégie from 2017 to 2019 (1.48, 2.44, and 4.03 nymphs/100 m^2^, respectively) and in Estrie in 2015 (1.27 nymphs/100 m^2^). Trends in average nymph densities at the provincial level were compatible with trends seen in the number of reported LD cases from 2015 to 2019 (Figure 4).

As administrative regions provide artificial boundaries, spatial interpolation of the data was carried out (Figure 5).

### 2.3. Statistical Analyses

The Poisson regression model showed a significant relationship between predicted nymph density and number of cases of LD (Z = 3.828; *p* < 0.001), with municipality ID as a random variable and the logarithm of the human population as an offset. The model estimated an increase of 1.38 cases for every unit increase in nymph density per 100 m^2^ per logarithmic unit of the population. The marginal R^2^ of the model was 0.021, and the conditional R^2^ was 0.831. Thus, while nymph density explained some of the variation in the model, the random effect, municipality ID, was crucial for following spatiotemporal trends.

Conditional predictions, which include the random effect, were calculated from the model and compared with the actual reported number of LD cases for each region of the study zone, across the study period (Figure 6). While predictions were generally able to capture inter-regional and inter-annual variations, predictions for Estrie between 2015 and 2019 did not show the increase in the number of human cases across this time period. However, when aggregated at the provincial level, predictions were able to follow trends (increase or decrease in risk) at this larger scale (Figure 7).

We compared the predicted number of human LS cases derived from our model and actual number of reported LD cases across the five-year study period at the district level (called *réseaux locaux de services* (RLS) in Québec) (Figure 6). The district scale was chosen for ease of visualisation. The model was able to estimate accurately the number of LD cases over a five-year period except for two RLS, one in Montérégie and the second in Lanaudière, where the number of LD cases was overestimated.

### 2.4. Risk Mapping with Modelling Results

Applying the criteria from Table 1 (see ‘Materials and Methods’ for more details) to the predictions from our fitted model allowed us to produce a classified LD risk map with the same risk categories as the Quebec provincial risk map produce by INSPQ for 2019 (Figure 8). Model predictions for municipalities in the highest risk category (“significant risk”) closely matched the spatial distribution of this risk category in the provincial risk map. The municipalities within the risk-level category “present” were predicted to be more widespread in the south of the province, and around Quebec City, than shown in the provincial risk map. Predictions for the models were most sensitive and specific for identifying municipalities with a “significant” risk level at 78.2% and 99.3%, respectively (Table 2).

## 3. Discussion

In this study, we investigated the ability of sentinel tick surveillance to capture spatiotemporal variation in human Lyme disease risk across the emergence zone for this disease in southern Quebec, Canada. We demonstrated that, even with limited sampling effort (21 sites sampled twice per year), sentinel surveillance reliably captured regional trends in LD risk over a large geographical area. Furthermore, we showed that risk maps generated from sentinel surveillance closely matched those derived from a more complex risk assessment based on multiple data sources. This first assessment of the application of sentinel surveillance in the context of emerging Lyme disease suggests that sentinel surveillance has the potential to provide a cost-effective approach for long-term monitoring of tick-borne disease risk over large geographic areas.

Our simple model, based on nymph densities alone as index of enzootic hazard, showed that an increase of 1 nymph per 100 m^2^ was associated with an increase of 1.38 human LD cases. In the literature, many studies have used infected nymph densities [16,17,28,29] as a measure of enzootic hazard, with similar positive associations with risk of LD. In contrast, our study was conducted on an area of emerging LD risk; nymph densities remain relatively low in active surveillance, and very few ticks test positive for *Borrelia*. Thus, our study supports the use of nymph density, as opposed to infected nymph density, as a representative enzootic hazard measure in areas where LD risk is emerging [9]. Some information bias may be included due to interpolation of the data, as these estimations were used in our models to predict LD risk. Interpolation across the study zone will hide some of the finer scale heterogenous presence of ticks [30]. While this bias is important to note, using the municipality as a random variable as allowed for predicted and observed human LD cases concord closely, thus limiting this source of bias. However, this leads to another limitation of our model: its applicability to other sentinel networks. Marginal R2 was much smaller than the conditional R2, showing that although nymph density can contribute to spatiotemporal LD risk predictions, inclusion of the study context is vital within the model. Ticks and LD are subject to fine-scale heterogeneity due to the effect of the geographical and ecological context [20,30]; thus, in surveillance of the risk, these factors remain an important piece of the puzzle.

We used seasonality models to correct nymph densities collected across the field season. This allows comparability of the enzootic hazard measure, otherwise limited due to tick phenology [31]. Development of seasonality models in the south of Quebec has made this adjustment possible [25], and public health authorities should evaluate the benefits of developing such models in their respective areas or use those which are already available in the literature [32,33,34]. We note that in previous studies conducted in eastern Canada, no such phenological adjustments had been carried out with active surveillance data to explain LD risk [9]. Furthermore, by interpolating sentinel surveillance data, our analyses cover a greater proportion of our study zone, with the benefit of maximizing the use of resource-intensive data collected on the field.

Some limits are important to consider with the use of seasonality models, for example, they may not be able to fully capture the inter-annual phenological variations in tick life cycle. This limit is minimised as seasonality models used more than one year of data during construction. The phenology may also slightly differ across the study zone; however, as we used a model developed for data in Québec, the differences should be minimal. To overcome this limit further, timing sampling visits during the same period would overcome the need to correct nymph densities.

Using model predictions, we were able to show that the model was able to match trends in human Lyme disease risk, both in time and space. Firstly, LD case predictions across different RSS through the study period capture the observed number of cases. Estrie was the region for which the model had the most difficulty in predicting observed case numbers. Estrie is also the region with the greatest number of human cases. However, LD incidence varies greatly across its area, and we had only two sentinel sites to capture this variation [27]. We suggest adding a sentinel site in the endemic district of Brome-Missiquoi [35], which could potentially allow better annual predictions of human cases in this region.

The temporal concordance with the predicted LD risk is less strong than the spatial component; sentinel surveillance did not always reliably track interannual variation in human cases within a region. However, the overall trend predicted from the model was consistent with human cases at the provincial scale. Sentinel surveillance can thus serve as a broad indicator of human case trends, allowing sentinel surveillance to forecast the approximate LD risk for the year and can foreshadow a surge in human cases.

Every year, the INSPQ produces municipal-scale risk map-based surveillance data collected the previous year. Using the same criteria, we classified our model predictions into risk levels to create a comparable risk map. The predictions were able to identify significant risk municipalities in the south of the provinces, with a sensitivity of 78.2%. While sensitivity was lower for “present” and “possible” risk levels, the predictions are better at identifying areas with an emerging risk of LD. The INSPQ criterion based on the number of human cases is not adapted to municipalities greater than 100,000 inhabitants and thus we could not include larger urban centers for our risk map [35]. There is a need to develop other case-based criteria for establishing risk levels for densely populated urban areas. Nonetheless, the ability of sentinel surveillance to track spatiotemporal changes in risk of LD may complement information derived from other sources (e.g., passive surveillance). An annual “Sentinel Surveillance Risk Indicator” can provide a reliable estimate of human risk across all municipalities, and is strongly correlated with the more rigorous, but more costly, measure of risk provided by integrated surveillance [36,37].

Our study shows that sentinel surveillance may provide a sustainable method to track spatiotemporal trends in LD risk over time. Such a network can support public health authorities in informing the public about LD risk within their region or municipality and this method could be extended or adapted to support Lyme risk assessment broader spatial scales, such as the national level with sentinel sites distributed among provinces or states. More generally, our study provides evidence of the utility of sentinel surveillance for monitoring temporal changes in emerging disease risk [36], overcoming strategic or logistical challenges associated with other methods of surveillance [9] while still providing reliable information on regional disease risk over extensive geographic areas.

## 4. Materials and Methods

### 4.1. Study Area

Quebec is located in the eastern past of Canada, sharing borders with the provinces of Ontario to the west, and New Brunswick and Newfoundland and Labrador to the east. To the south, Quebec neighbors the states of New York, Vermont, New Hampshire, and Maine. Its area totals over 1,350,000 km^2^, divided in 17 health regions (*Régions Socio-Sanitaires*: RSS), each with their own regional public health directorate (*Direction de Santé Publique*: DSP). Of the 17 RSS which make up the province, 10 are areas of key scientific interest for the emergence of Lyme disease, due to their more southerly geographical positions; this study will focus on these 10 RSS (Figure 1).

### 4.2. Sentinel Surveillance in Quebec

Since 2010, the INSPQ, the Public Health Agency of Canada (PHAC) and the University of Montreal have jointly coordinated active surveillance in southern Quebec [12]. From 2015 onwards, a network of sentinel sites was designed by the group of experts on tick-borne diseases (Groupe d’experts sur les maladies transmises par les tiques), a panel formed by scientific and medical advisors, epidemiologists, public health officials, and laboratory experts specialized in vector-borne diseases. Two sites were chosen per region, except in the region of Montreal where three sites were selected due to high population density, for a total of 21 sentinel sites (Figure 1). The sites were placed in provincial or regional parks, characterized by suitable deciduous forest habitat for the establishment of tick populations, and located in geographically distinct areas of each RSS.

Sentinel sites were sampled twice during the summer activity period of nymphal *I. scapularis* ticks in southern Quebec (May–August): Once in late May to early June, followed by a second visit between July and mid-August. From 2018, two sites considered by public health authorities to have a well-established tick population due to the large number of ticks collected (one site in Montérégie and another in Estrie) were sampled only once per year, to allow for allocation of sampling resources to other sites where the risk was evolving. Exceptionally, some sites were only visited once per year due to logistical constraints, e.g., park closure. Finally, in 2016 and 2017 some sites were visited three times due to other research projects occurring concurrently.

A standardized drag sampling protocol was carried out at each site. Each site is sampled by two team field technicians, each dragging a 1 m^2^ piece of flannel cloth horizontally on the ground along two transects: One along the vegetation at the edge of a public nature trail, and the second parallel transect in the forest 25 m from the trail. Each team member sampled 1000 m^2^, for a total sample area of 2000 m^2^ per site. During sampling, the presence of ticks on the flannel was checked every 25 m. Ticks were removed with tweezers, placed in tubes filled with 70% ethanol, and sent to the Quebec Public Health Laboratory (Laboratoire de Santé Publique du Québec—LSPQ) for species identification and pathogen testing.

### 4.3. Density of Questing Nymphs

To represent enzootic hazard, nymph densities (nymph/100 m^2^ of surface area dragged) were calculated from sentinel site visits. Nymph density was used because nymphs have been shown to represent the greatest hazard to human health due to their small size, in comparison to the adult stage, their possibility of being infected, in comparison with the larval stage, and their activity peak during summer months [14].

Surveillance activities are carried out at different times during the summer, and tick phenology (seasonal activity) varies across this period [31], resulting in predictable variation in nymph densities observed during early and late visits to the same site. We therefore used seasonality models of *I. scapularis* phenology in the south of Quebec developed by Dumas et al. [25] to correct the raw nymph densities, using a reference date of 15 June, corresponding to the expected peak of nymph activity. We used these corrected nymph densities to compute the mean density per site per year across the study period.

### 4.4. Notifiable Disease Surveillance System

Lyme disease has been a reportable disease in Quebec since 2003 [35]. Any suspected or diagnosed cases of LD, based on a standardized case definition, must be reported to the notifiable diseases database, kept at the regional level. The case definition of LD in Canada is based on clinical manifestations, likely location of acquisition, and the use of diagnostic tests as described by the federal public health guidelines [38]. Once the cases are reported by physicians, each DSP is responsible for conducting a public health investigation of each case in their region. Thus, regional databases belong to each of the DSPs. For the purposes of this study, we requested access to the human case data at the municipal level for the 10 RSS covered by the Quebec active tick surveillance system (Figure 1), between 2015 and 2019.

### 4.5. Statistical Analyses

A Poisson mixed model was constructed to evaluate the relationship between average nymph density, the index enzootic hazard generated from sentinel surveillance, and human LD risk, measured as the annual number of human cases at the municipal level with human population as an offset (see below). We estimated annual hazard measures for each municipality by spatial interpolation of average nymph densities measured across the network of sentinel sites for each year of the study period. Interpolation was carried out using a Kernel density estimation in QGIS version 3.18 Zurich [39]. A distance of 80 km was used as the radius of interpolation, as correlograms revealed spatial dependency of active surveillance data up to this distance [40].

Statistical models were fitted using the lme4 package in R version 3.6.2 [41]. In addition to the risk (dependent variable) and the enzootic hazard (independent variable), the log of the human population was used as an offset. We used population data from Statistics Canada’s Census of Population 2016 [24]. The municipality code was added as a random effect to account for repeated measures taken across the five years of the study.

Conditional model predictions of the annual number of human cases in each municipality were compared graphically with observed number of cases reported through the Notifiable Disease Surveillance System.

### 4.6. Risk Mapping with Modelling Results

Lastly, we used our predictions to simulate the Quebec LD risk map using the risk criteria for human cases established by the INSPQ. Yearly, the INSPQ produces an LD risk map at the municipal scale [26]. The risk map uses three levels of risk: Significant, present, and possible. As the INSPQ also uses passive surveillance data in addition to active surveillance data, we adapted the 2019 INSPQ criteria so these could be used for our data set, which also uses data derived from active surveillance (Table 1). Applying these criteria to our model predictions, we constructed a risk map for 2019 and compared it to the risk map created by the INSPQ for the same year. The sensitivity and specificity of the classification, in comparison with the provincial risk map, are presented for the “significant” and “present” risk categories.

## Figures and Tables

**Figure 1 pathogens-11-00531-f001:**
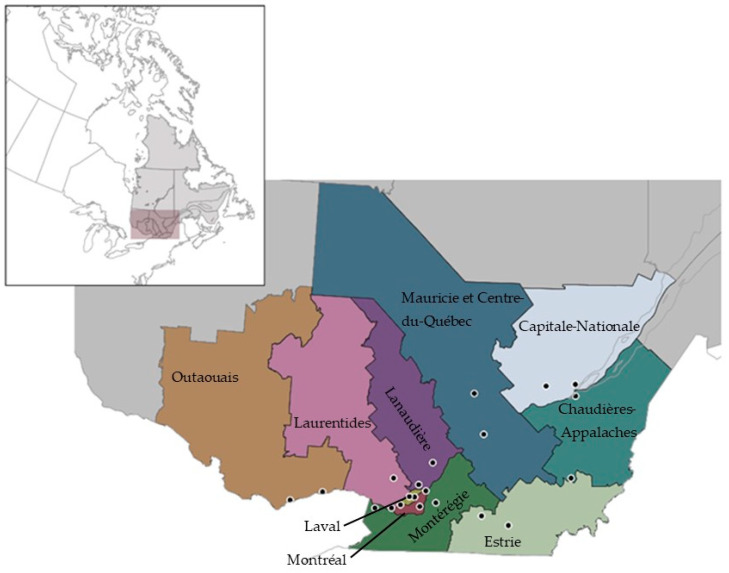
Health region (*Régions socio-sanitaires*: RSS) part of Quebec’s LD active surveillance system, with sentinel site distribution across the study zone; two sites were chosen per region in the south of the province with the exception of Montreal which had three sites.

**Figure 2 pathogens-11-00531-f002:**
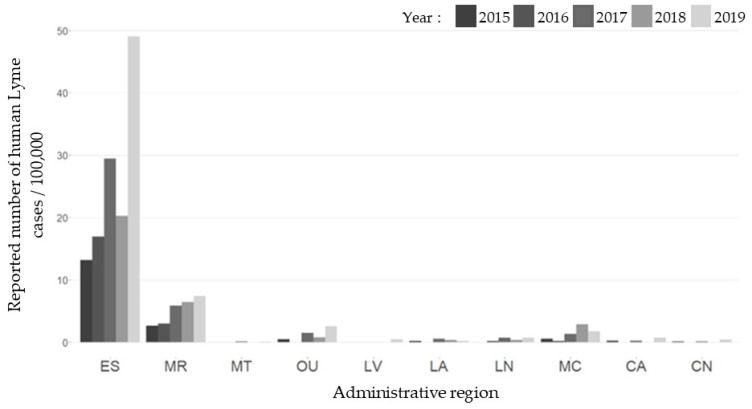
Reported number of human Lyme disease cases per 100,000 population for each region from 2015 to 2019. Legend: ES: Estrie; MR: Montérégie; MT: Montreal; OU: Outaouais; LV: Laval; LA: Laurentides; LN: Lanaudière; MC: Mauricie et Centre-du-Québec; CA: Chaudière-Appalaches; CN: Capitale-Nationale.

**Figure 3 pathogens-11-00531-f003:**
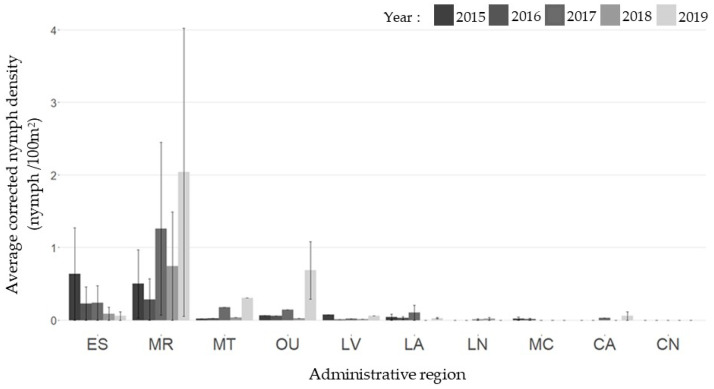
Average nymph densities per year in each MSSS, corrected for the 15th of June using seasonality models [25]. Legend: ES: Estrie; MR: Montérégie; MT: Montreal; OU: Outaouais; LV: Laval; LA: Laurentides; LN: Lanaudière; MC: Mauricie et Centre-du-Québec; CA: Chaudière-Appalaches; CN: Capitale-Nationale.

**Figure 4 pathogens-11-00531-f004:**
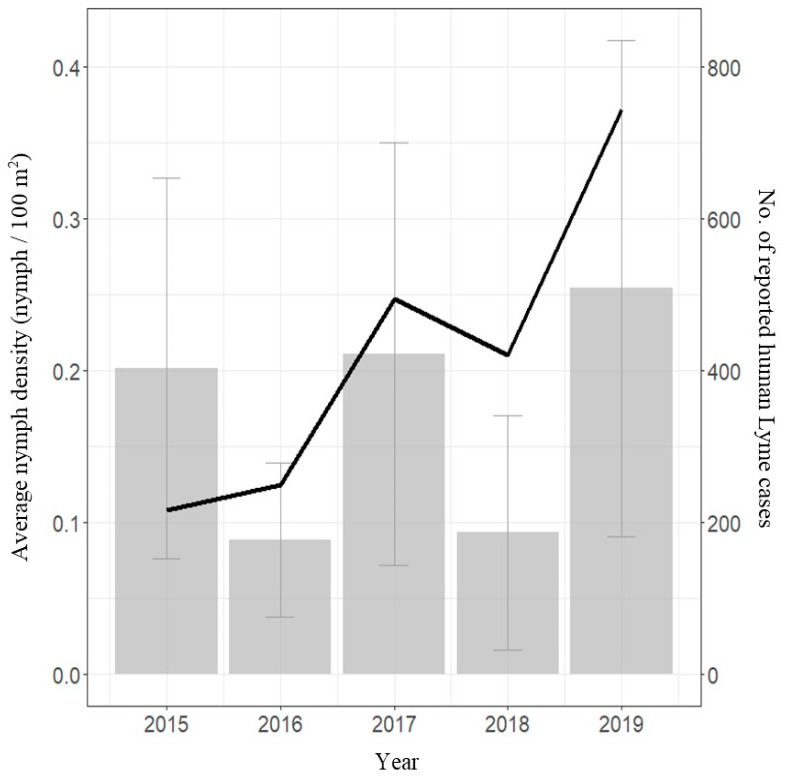
Diagram comparing average nymph densities at the provincial level from 2015 to 2019 (bars) with confidence intervals, with observed number of human cases (line).

**Figure 5 pathogens-11-00531-f005:**
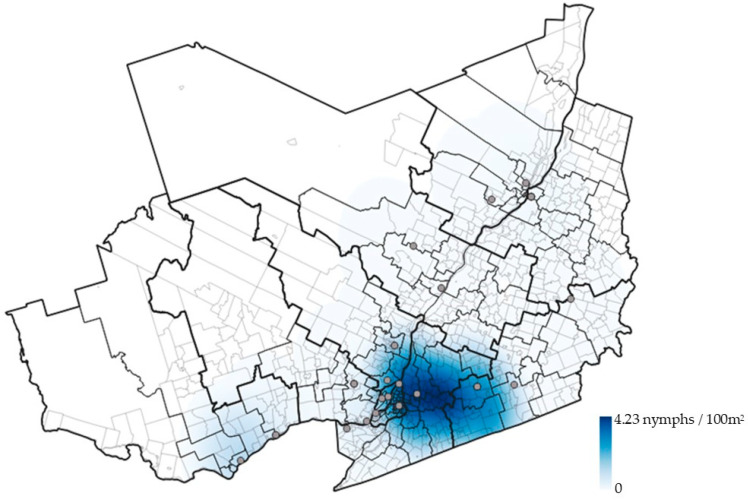
Interpolated enzootic risk for Lyme disease across the study zone (southern Quebec) for 2019; enzootic risk is derived from sentinel surveillance and interpolated using Kernel density estimation and reported as density of nymphs (DON) (nymph/100 m^2^). Sentinel site locations are represented using grey points.

**Figure 6 pathogens-11-00531-f006:**
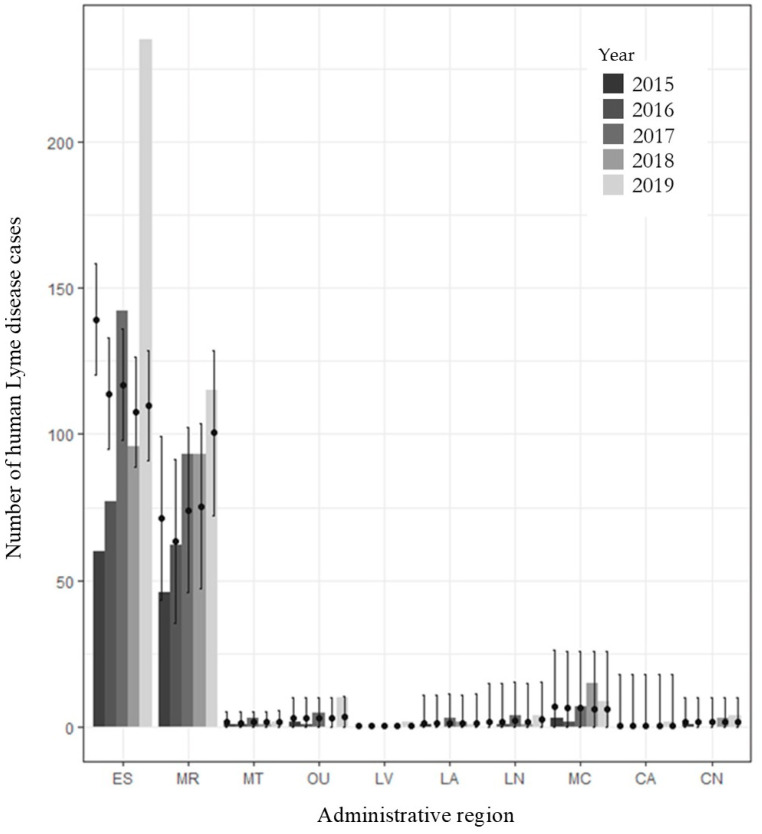
Comparison of model predictions of human LD cases (black dots) with confidence intervals, with observed number of human cases (bars) at the regional level across the study period from 2015 to 2019. Legend: ES: Estrie; MR: Montérégie; MT: Montréal; OU: Outaouais; LV: Laval; LA: Laurentides; LN: Lanaudière; MC: Mauricie et Centre-du-Québec; CA: Chaudière-Appalaches; CN: Capitale-Nationale.

**Figure 7 pathogens-11-00531-f007:**
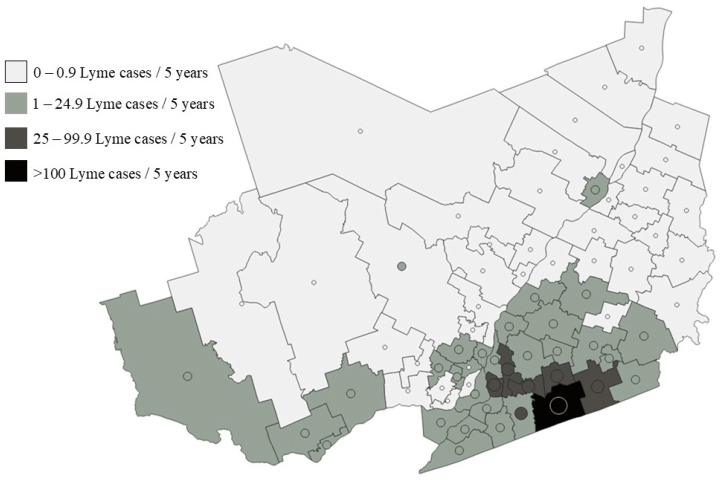
Map comparing predicted number human Lyme disease (LD) cases over the study period (2015–2019) with observed number of human LD cases, at the district level (called *réseaux locaux de services* (RLS) in Québec). The district scale was chosen for ease of visualisation. The district color represents the predicted number of LD cases whilst the superimposed circles represent the observed number of cases. The diameter of circles is scaled according to the logarithm of the number of human cases.

**Figure 8 pathogens-11-00531-f008:**
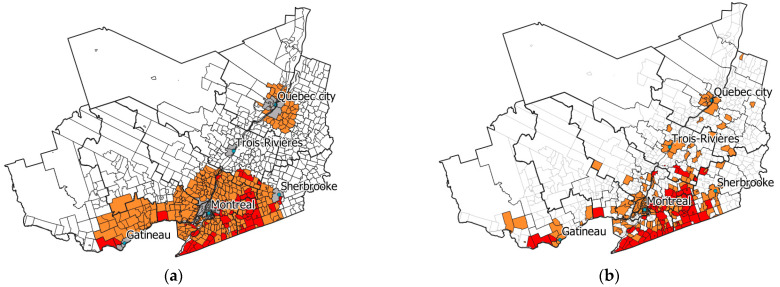
Maps comparing municipal LD risk levels scaled for 2019 as derived from (**a**) sentinel surveillance model predictions (municipalities with >100,000 excluded) and (**b**) the INSPQ [27]. Legend: Red: Significant risk; yellow: Present risk; white: Possible risk. Criteria for determining risk levels are presented in Table 1.

**Table 1 pathogens-11-00531-t001:** Criteria used to classify risk level of LD at the municipal level in Quebec, adapted from the INSPQ [26].

Risk Level	Criteria
Significant	At least three cases of Lyme locally acquired in the last five years
Present	A nymph density of at least 0.05 ticks/100 m^2^ Two cases of LD locally acquired in the last five years
Possible	Municipalities which do not meet “significant” or “present” criteria

**Table 2 pathogens-11-00531-t002:** Sensitivity and specificity of predictions from models to classify municipal-level risk of Lyme disease in the south of Québec.

Risk Level	Sensivity (%)	Specificity (%)
Present	71.9	65.4
Significant	79.7	99.3

## Data Availability

Human LD cases data from the Notifiable Disease Surveillance System belong to the Direction de santé publique at the regional (RSS) level. Tick density data from the sentinel surveillance network belong to the INSPQ and the Université de Montréal.

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
