# Peer review of "Sentinel Surveillance Contributes to Tracking Lyme Disease Spatiotemporal Risk Trends in Southern Quebec, Canada"

_pathogens, 2022, doi:10.3390/pathogens11050531_

Round 1

Reviewer 1 Report

Structure of paper and discussion part are at a high level. I have some comments, which should be take in consideration:

  • Affiliations are in French, please change them to English;
  • In text, there are also some phrases in French (for example: in Line 96-97 Introduction; line 32 - material and methods (in 4.2. Sentinel (...);
  •  Abstract is written chaotically, I recommended using the mdpi template and change it to structured abstract - then it will be much clearer;
  • Line 95 - should be Ixodes scapularis, not I. scapularis (first time in text)
  • Please, do not use Italics, when it is not necessary (Regions), such as: Line 12 (Results), Line 17 (2.3. Statistical analyses), Line 19 (2.3. Statistical analyses), etc.
  • Please, tell me what is the idea of table 1?
  • All figures and tables should be fully understandable without looking for data in the text - please take this into account (no abbreviations, adding numerical values (Figure 3), Figure 4-Line 15 human Lyme Disease cases, etc.

Author Response

Thank you very much for taking the time to review this manuscript

Responses to comments are in the attached document. 

Kind regards, 

Camille 

Reviewer 2 Report

Manuscript ID: pathogens-1693876

Sentinel surveillance contributes to tracking Lyme disease spatiotemporal risk trends in Southern Quebec, Canada

The manuscript written by Guillot and co-workers described the potential benefits and disadvantages of using the setinel surveillance contributes to tracking Lyme disease spatiotemporal risk trends in Southern Quebec, Canada. The manuscript is well-written, I have only few question.

Did you check if captured ticks were infected by Borrelia? I think it is important to calculate the prevalence of Borrelia in ticks and also this result should be include in the model.

Please add the hypothesis in the end of Introduction section.

I propose to use PCA analysis to demonstrate the results.

Author Response

Thank you very much for taking the time to review this manuscript.

Responses to comments are in the attached document. 

Kind regards, 

Camille 

Round 2

Reviewer 2 Report

I agree with  the authors responses. I accept the manuscript with present form.